∂ | **Open Peer Review** | Clinical Microbiology | Research Article

# Global, regional, and national burden of HIV and other sexually transmitted infections among women of childbearing age from 1990 to 2021

Xiaoyu Zhang,[1,2] Chenglong Hu,[1,2] Yao Liang,[1,2] Wanguo Dong,[1,2] Jian Gao,[1,2] Yu Ji,[1,2] Chang Cao,[1,2] Wei Shi,[1,2] Shuaijie Zhu,[1,2] Heng Guo,[1,2] Tianfeng Hua,[1,2] Hui Li,[1,2] Min Yang[1,2]

**ABSTRACT** Sexually transmitted infections (STIs) represent a significant public health burden, particularly among women of childbearing age. This study aims to analyze the global, regional, and national age-standardized rates of incidence, prevalence, mortality, and disability-adjusted life years (DALYs) for STIs among women aged 15–49 from 1990 to 2021. Data were sourced from the Global Burden of Disease 2021 database, and age-standardized rates were calculated using direct standardization methods. Temporal trends were evaluated through average annual percentage change (AAPC) via jointpoint regression analysis. In 2021, compared with other STIs, HIV/AIDS had the highest age-standardized mortality rate (12.98 per 100,000; 95% CI: 10.04 to 16.84) and DALY rate (829.89; 95% CI: 658.73 to 1,056.91). Trichomoniasis had the highest incidence (6,709.73; 95% CI: 3,676.25 to 10,839.25), and genital herpes had the highest prevalence (17,137.09; 95% CI: 13,485.32–21,121.75). From 1990 to 2021, trichomoniasis showed the sharpest increase in incidence (AAPC: 0.27 [95% CI: 0.25 to 0.30]), while HIV/AIDS had the greatest rise in prevalence (AAPC: 3.50 [3.35 to 3.65]), mortality (AAPC: 1.49 [0.97 to 2.02]), and DALYs (AAPC: 1.52 [1.02 to 2.01]). In contrast, gonococcal infection exhibited the steepest declines in prevalence (AAPC: −0.46 [−0.49 to −0.43]) and mortality (AAPC: −1.16 [−1.34 to −0.97]), while syphilis had the largest decrease in DALYs (AAPC: −1.14 [−1.32 to −0.96]). Regional and national disparities were evident across all metrics. These findings highlight the ongoing and uneven burden of STIs and the need for tailored screening and prevention strategies.

**IMPORTANCE** Sexually transmitted infections (STIs) pose a significant public health challenge, particularly among women of childbearing age, with substantial impacts on individual health and societal wellbeing. This study provides a comprehensive analysis of the global, regional, and national age-standardized rates of incidence, prevalence, mortality, and disability-adjusted life years (DALYs) for STIs among women aged 15–49 over the past three decades. The findings reveal alarming trends, with HIV/AIDS topping the list in terms of mortality and DALYs and trichomoniasis and genital herpes showing high incidence and prevalence rates. These data highlight the urgent need for targeted screening and preventive interventions to address the disparities in STI burden across regions and countries. By understanding the trends and patterns of STIs, policymakers and healthcare providers can develop effective strategies to reduce the transmission and impact of these infections among women of reproductive age, thereby improving global public health outcomes.

**KEYWORDS** sexually transmitted diseases, HIV, women of childbearing age

**Peer Reviewers** Filipe Cerqueira, University of Alabama at Birmingham, Birmingham, Alabama, USA; Anatasia Weiland, University of California Irvine, Irvine, California, USA

Address correspondence to Min Yang, yangmin@ahmu.edu.cn.

The authors declare no conflict of interest.

See the funding table on p. 11.

STIs remain a significant global public health concern, encompassing a wide range of diseases such as HIV and syphilis, which are primarily transmitted through sexual contact and, in some cases, through vertical transmission routes (1–3). The burden of STIs is particularly severe among individuals under 49 years of age (4–6). Women of childbearing age, especially those who are sexually active, are at increased risk of these infections due to their unique anatomical features. The vaginal mucosa is thin and easily penetrated by pathogens, and the female reproductive tract allows for easier upward spread of infections, increasing the risk of complications such as pelvic inflammatory disease and infertility (7). In addition to the immediate health risks, STIs can lead to severe consequences, including infertility, preterm birth, and the vertical transmission of infections (8–10). Despite the recognized importance of this issue, there is a lack of comprehensive assessment regarding the global and regional burden of STIs among women of childbearing age. This gap in understanding hinders effective disease control efforts. Furthermore, while HIV/AIDS has received substantial attention, other STIs may not be as prominently addressed (11, 12).

The Global Burden of Disease, Injuries, and Risk Factors Study 2021 (GBD 2021) is a comprehensive database widely used in epidemiological research to assess the burden of disease, age-standardized incidence rates, and trends over time (13, 14).

In this study, we extracted data on STIs, including HIV/AIDS, syphilis, chlamydial infection, gonococcal infection, genital herpes, and trichomoniasis, from the GBD 2021 database. We analyzed age-standardized rates of incidence, prevalence, mortality, and disability-adjusted life years (DALYs) and examined temporal trends in women of childbearing age from 1990 to 2021 at global, regional, and national levels.

## MATERIALS AND METHODS

### Data acquisition

GBD 2021 provides a comprehensive evaluation of the health impacts associated with 369 diseases, injuries, and disabilities, along with an analysis of 88 risk factors across 204 countries and territories over recent decades. At the regional level, the GBD 2021 categorizes regions into five quintiles based on the sociodemographic index (SDI) values. This aggregate measure reflects social and economic factors influencing health outcomes across different geographical areas. The database also identifies 21 geographically proximate regions for more localized analysis (5). The statistical methods and standardization techniques used in GBD 2021 have been previously described in other studies (5, 15). In this study, we focused on data related to STIs, including HIV/AIDS, syphilis, chlamydial infection, gonococcal infection, genital herpes, and trichomoniasis. These rates are reported per 100,000 individuals. Our analysis specifically targeted females aged 15–49 years, a group categorized by the World Health Organization (WHO) as women of childbearing age [https://www.who.int/data/gho/indicator-metadata-registry/imr-details/women-of-reproductive-age-(15-49-years)-population-(thousands)]. Data on incidence, prevalence, mortality, and DALYs were extracted from GBD 2021, covering the years 1990 to 2021. The data set includes global, regional, and national data from the five SDI regions, 21 GBD regions, and 204 countries and territories.

### Data analysis

To enhance data comparability and exclude the potential influence of population structure, age standardization was performed. In this study, we applied a direct standardization method using the age structure of the standard world population. This method adjusts for population differences to analyze age-standardized rates of the incidence, prevalence, mortality, and DALYs. The estimation methods have been described in previous studies (16–18). To evaluate trends in disease burden over time, we calculated the AAPC between 1990 and 2021 using jointpoint regression models. An increasing trend was identified when both the estimated AAPC and its lower 95%

confidence interval (CI) exceeded 0. Conversely, a decreasing trend was identified when both the estimated AAPC and its upper 95% CI were below 0. If the AAPC did not meet either criterion, the trend was considered stable throughout the study period. Data analysis was performed using R software (version 4.3.2).

## RESULTS

### HIV/AIDS

In 2021, the estimated age-standardized rates for incidence, prevalence, mortality, and DALYs in women of childbearing age were as follows: 36.99 (32.06 to 43.14), 804.77 (759.54 to 858.92), 12.98 (10.04 to 16.84), and 829.89 (658.73 to 1,056.91), respectively (Table S1). Between 1990 and 2021, the AAPC in the age-standardized incidence rate was −1.88 (−2.17 to −1.59), suggesting a decline in incidence over the period. Conversely, the AAPCs for prevalence, mortality, and DALYs were 3.50 (3.35 to 3.65), 1.49 (0.97 to 2.02), and 1.52 (1.02 to 2.01), respectively, reflecting increasing trends (Table S2).

At the regional level, southern Sub-Saharan Africa had the highest rates across all metrics in 2021. The values were as follows: 575.79 (439.93 to 730.32), 22,745.44 (21,536.92 to 24,099.54), 230.12 (195.75 to 278.99), and 15,578.14 (13,430.42 to 18,465.43), respectively, for incidence, prevalence, mortality, and DALYs (Table S1). From 1990 to 2021, Oceania experienced the largest increases in age-standardized incidence and prevalence rates, with AAPCs of 11.60 (10.90 to 12.30) for incidence and 16.90 (16.29 to 17.52) for prevalence. South Asia showed the greatest increases in mortality and DALY rates, with AAPCs of 18.48 (15.72 to 21.31) for mortality and 16.70 (14.59 to 18.84) for DALYs.

Lesotho had the highest national rates for HIV/AIDS incidence, prevalence, mortality, and DALYs, with values of 1,324.93 (745.53 to 2,131.74) for incidence, 32,032.26 (27,208.12 to 36,481.06) for prevalence, 514.39 (278.78 to 837.44) for mortality, and 32,047.85 (18,615.97 to 50,424.95) for DALYs. From 1990 to 2021, Pakistan experienced the largest increases across all metrics, with AAPCs of 30.36 (25.27 to 35.67) for incidence, 32.88 (30.99 to 34.81) for prevalence, 39.44 (37.12 to 41.79) for mortality, and 38.21 (36.15 to 40.3) for DALYs. Conversely, Burundi saw the most significant drop in the incidence rate (AAPC: −14.38 [−14.98 to −13.78]), while Burkina Faso had the most notable declines in the other three metrics, with AAPCs of −5.96 (−6.09 to −5.83) for prevalence, −8.21 (−8.86 to −7.55) for mortality, and −8.13 (−8.75 to −7.50) for DALYs (Table S3; Fig. 1).

### Syphilis

In 2021, the global age-standardized rates for syphilis were as follows: incidence 112.28 (57.32 to 190.60), prevalence 1,121.17 (684.72 to 1,721.54), mortality 0.01 (0.01 to 0.02), and DALYs 2.10 (1.55 to 2.85) (Table S1). Over the period from 1990 to 2021, the age-standardized incidence and prevalence rates remained relatively stable, with AAPCs of 0.26 (−0.10 to 0.63) and 0.09 (−0.03 to 0.21), respectively. In contrast, both age-standardized DALYs and mortality rates declined significantly, with AAPCs of −1.14 (−1.32 to −0.96) for DALYs and −0.70 (−0.91 to −0.48) for mortality (Table S2).

At the regional level in 2021, central Sub-Saharan Africa exhibited the highest rates of incidence and prevalence, with values of 1,210.05 (636.64 to 1,997.65) and 4,689.77 (2,830.28 to 7,249.79), respectively. Eastern Sub-Saharan Africa had the highest mortality and DALY rates at 0.059 (0.030 to 0.127) and 8.41 (5.70 to 12.95), respectively (Table S1). From 1990 to 2021, tropical Latin America saw the most significant increase in age-standardized incidence and prevalence, with AAPCs of 1.27 (0.91 to 1.64), and 1.26 (0.96 to 1.56), respectively. The Caribbean showed the largest increase in age-standardized mortality and DALYs, with AAPCs of 1.19 (0.55 to 1.83) for mortality and 0.75 (0.32 to 1.18) for DALYs (Table S2).

In 2021, the highest age-standardized incidence rate was observed in Equatorial Guinea (1,506.26 [792.30 to 2,490.52]), the highest prevalence in Liberia (6,274.26 [3,733.09 to 9,605.15]), the highest mortality in South Sudan (0.096 [0.029 to 0.276]), and

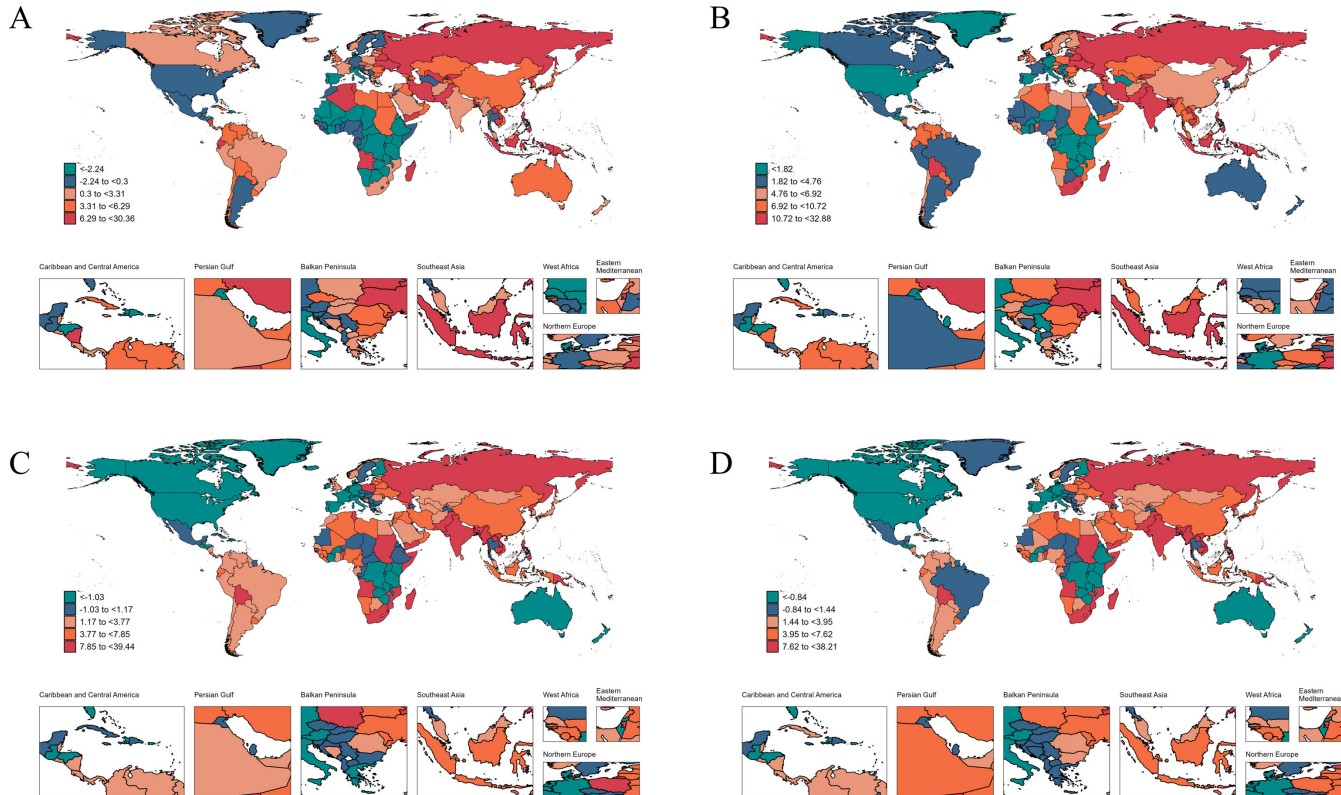

**FIG 1** The average annual percent change in age-standardized incidence (A), prevalence (B), mortality (C), and DALYs (D) rates of HIV/AIDS from 1990 to 2021.

the highest DALY rate in the Central African Republic (16.57 [9.43 to 26.30]) (Table S3). Between 1990 and 2021, the countries with the largest increases in age-standardized incidence, prevalence, mortality, and DALY rates were Brazil, Mongolia, Kuwait, and Dominica, respectively. The corresponding AAPCs were as follows: incidence 1.29 (0.94 to 1.65), prevalence 3.37 (1.99 to 4.77), mortality 13.06 (8.49 to 17.83), and DALYs 4.29 (4.06 to 4.53). In contrast, the most notable decreases in age-standardized incidence, prevalence, mortality, and DALY rates were observed in Malawi, Mozambique, the Northern Mariana Islands, and Armenia, respectively. The corresponding AAPCs were as follows: incidence −2.54 (−2.85 to −2.23), prevalence −3.52 (−5.42 to −1.59), mortality −7.18 (−9.01 to −5.30), and DALYs −6.02 (−7.85 to −4.15) (Table S3; Fig. 2).

## Chlamydial infection

In 2021, the global age-standardized rates of incidence, prevalence, mortality, and DALYs in women of childbearing age were as follows: incidence 5,179.06 (2,938.63 to 8,417.46), prevalence 4,570.25 (2,705.08 to 7,271.25), mortality 0.025 (0.015 to 0.037), and DALYs 4.58 (3.09 to 6.76) (Table S1). Between 1990 and 2021, the age-standardized incidence and prevalence showed no significant change (AAPCs: 0.10 [−0.11 to 0.30]; 0.10 [−0.09 to 0.30]), while the mortality and DALY rates declined (AAPCs: −1.12 [−1.30 to −0.95]; −0.27 [−0.35 to −0.20]) (Table S2).

In 2021, among the 21 GBD regions, Oceania had the highest age-standardized rates of incidence 13,457.05 (7,805.12 to 21,350.70) and prevalence 11,416.49 (6,706.78 to 17,990.07) (Table S1). eastern Sub-Saharan Africa exhibited the highest rates of mortality 0.112 (0.055 to 0.237) and DALYs 11.00 (6.54 to 19.81) (Table S1). From 1990 to 2021, high-income North America saw the most pronounced increases in age-standardized incidence (AAPC: 0.59 [0.38 to 0.81]) and prevalence (AAPC: 0.37 [0.16 to 0.57]). Some regions, including the Caribbean and tropical Latin America, saw increases, with the most

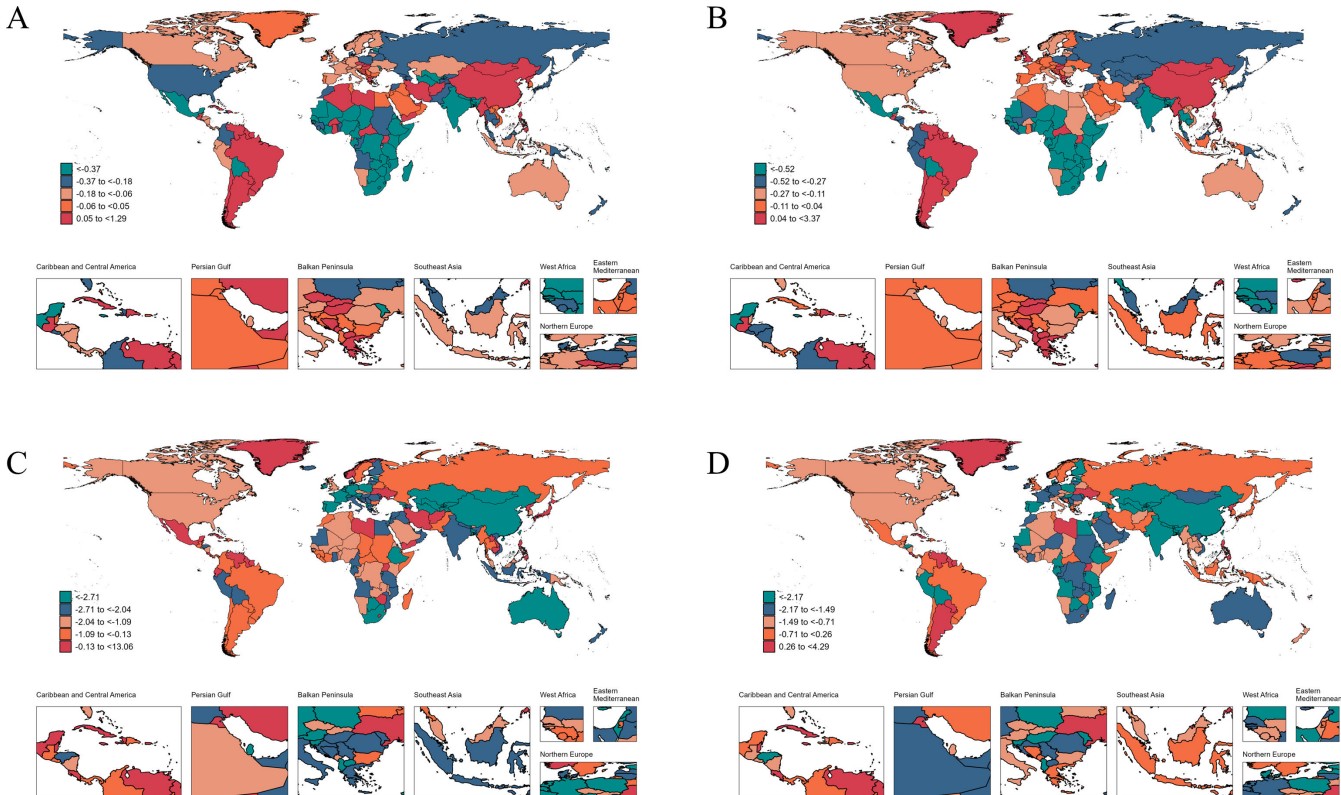

**FIG 2** The average annual percent change in age-standardized incidence (A), prevalence (B), mortality (C), and DALYs (D) rates of syphilis from 1990 to 2021.

significant rise in mortality rates observed in the Caribbean (AAPC: 1.20 [0.55 to 1.85]) and in DALY rates in tropical Latin America (AAPC: 1.85 [1.67 to 2.02]) (Table S2).

At the national level in 2021, Fiji reported the highest age-standardized incidence and prevalence rates of chlamydial infection, at 17,317.70 (10,149.13 to 27,142.43) for incidence and 14,617.66 (8,645.71 to 22,791.91) for prevalence. South Sudan recorded the highest mortality and DALY rates, at 0.180 (0.060 to 0.520) for mortality and 15.65 (6.58 to 37.42) for DALYs. Between 1990 and 2021, the United States experienced the most significant increase in age-standardized incidence rates (AAPC: 0.63 [0.44 to 0.81]), while the United Kingdom led in the growth of prevalence rates (AAPC: 0.85 [0.74 to 0.96]). Kuwait showed the largest increase in mortality rates (AAPC: 11.72 [6.94 to 16.71]), and Brazil demonstrated the most pronounced increase in DALY rates (AAPC: 1.9 [1.72 to 2.09]). Conversely, the steepest declines were observed, with Senegal showing declines for both incidence (AAPC: −1.23 [−1.31 to −1.15]) and prevalence (AAPC: −1.31 [−1.40 to −1.23]), the Northern Mariana Islands for mortality (AAPC: −7.17 [−9.09 to −5.20]), and Ethiopia for DALYs (AAPC: −3.15 [−3.34 to −2.97]) (Table S4; Fig. 3).

## Gonococcal infection

In 2021, the global age-standardized rates of gonococcal infection in women of child-bearing age were estimated as follows: incidence 1,430.11 (857.81 to 2,239.55), prevalence 1,008.52 (625.98 to 1,548.20), mortality 0.009 (0.005 to 0.013), and DALYs 1.24 (0.85 to 1.84) (Table S1). Between 1990 and 2021, all metrics exhibited declining trends, as reflected in their AAPCs: −0.46 (−0.49 to −0.42) for incidence, −0.46 (−0.49 to −0.43) for prevalence, −1.16 (−1.34 to −0.97) for mortality, and −0.75 (−0.84 to −0.67) for DALYs (Table S2).

In 2021, Oceania recorded the highest age-standardized rates for both incidence (8,770.75 [4,521.47 to 15,285.40]) and prevalence (5,890.13 [3,057.80 to 10,233.65]). Eastern Sub-Saharan Africa had the highest mortality rates (0.038 [0.019 to 0.081]) and

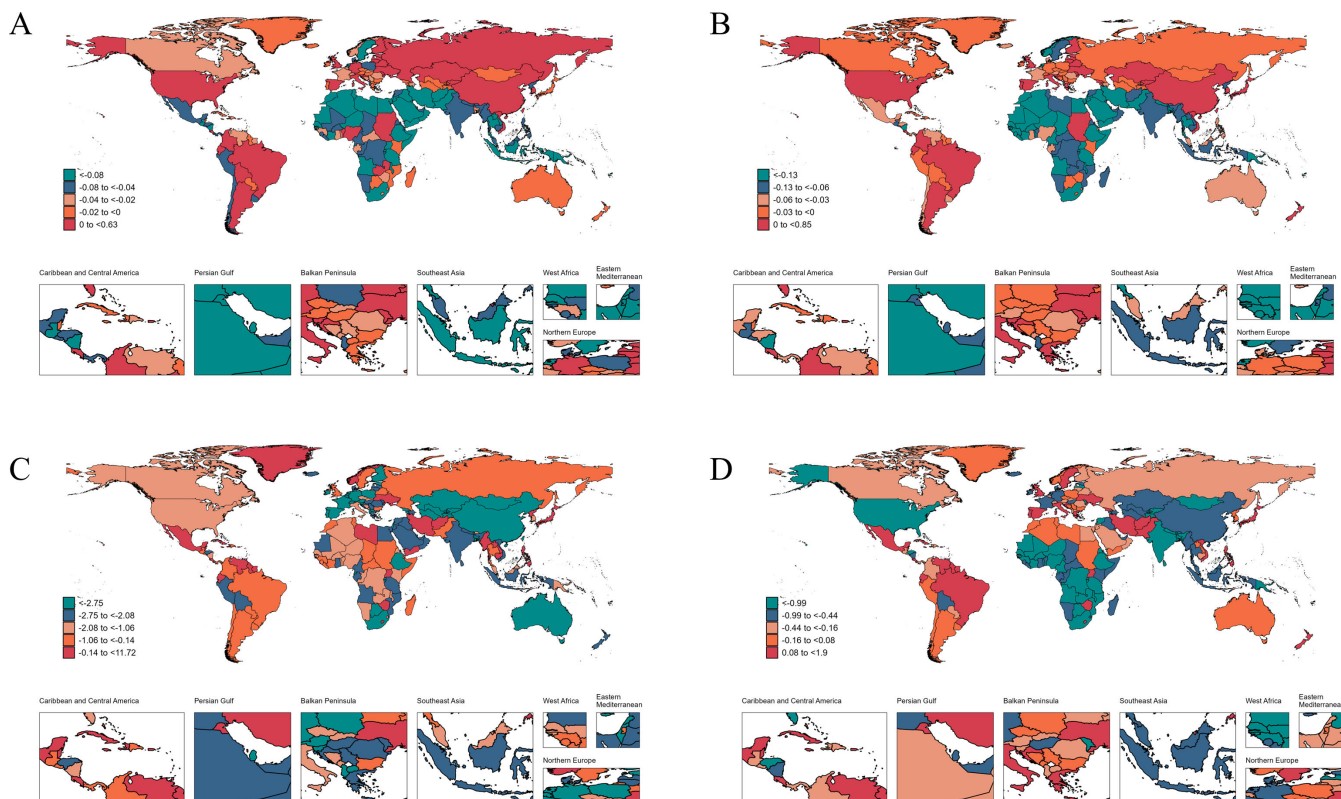

**FIG 3** The average annual percent change in age-standardized incidence (A), prevalence (B), mortality (C), and DALY (D) rates of chlamydial infection from 1990 to 2021.

DALY rate (3.72 [2.23 to 6.71]) (Table S1). From 1990 to 2021, Oceania reported the largest increases in incidence (AAPC: 0.34 [0.32 to 0.36]) and prevalence (AAPC: 0.33 [0.30 to 0.35]). The Caribbean exhibited the most pronounced rise in mortality (AAPC: 1.18 [0.55 to 1.83]), while tropical Latin America showed the highest growth in DALY rates (AAPC: 1.20 [1.01 to 1.39]) (Table S2).

Across nations, the highest age-standardized rates for gonococcal infection in 2021 were observed in Papua New Guinea for both incidence 10,495.59 (5,402.89 to 18,253.83) and prevalence 7,041.67 (3,647.84 to 12,214.27). South Sudan recorded the highest rates for mortality and DALYs, 0.063 (0.019 to 0.179) and 5.64 (2.54 to 13.34), respectively. From 1990 to 2021, the most significant growth in age-standardized rates was recorded in Norway for incidence (AAPC: 0.41 [0.38 to 0.45]), Sri Lanka for prevalence (AAPC: 0.37 [0.27 to 0.47]), Kuwait for mortality (AAPC: 13.12 [8.54 to 17.89]), and the Philippines for DALYs increase (AAPC: 2.1 [1.84 to 2.36]). Conversely, the steepest declines were observed in Tunisia for incidence (AAPC: −0.98 [−1.13 to −0.83]), Senegal for prevalence (AAPC: −1.29 [−1.5 to −1.08]), the Northern Mariana Islands for mortality (AAPC: −7.11 [−8.84 to −5.35]), and the Maldives for DALYs (AAPC: −3.43 [−3.74 to −3.12]) (Table S4; Fig. 4).

## Trichomoniasis

Globally, in 2021, the estimated age-standardized rates of trichomoniasis were 6,709.73 (3,676.25 to 10,839.25) for incidence, 5,552.21 (2,918.06 to 9,234.17) for prevalence and 10.54 (3.51 to 24.28) for DALYs (Table S1). Over the period from 1990 to 2021, all three metrics exhibited upward trends, with AAPCs of 0.27 (0.25 to 0.30) for incidence, 0.24 (0.19 to 0.28) for prevalence, and 0.24 (0.19 to 0.28) for DALYs (Table S2).

Regionally, southern Sub-Saharan Africa reported the highest age-standardized rates in 2021, with incidence at 21,438.37 (12,145.47 to 33,847.43), prevalence at 19,947.23

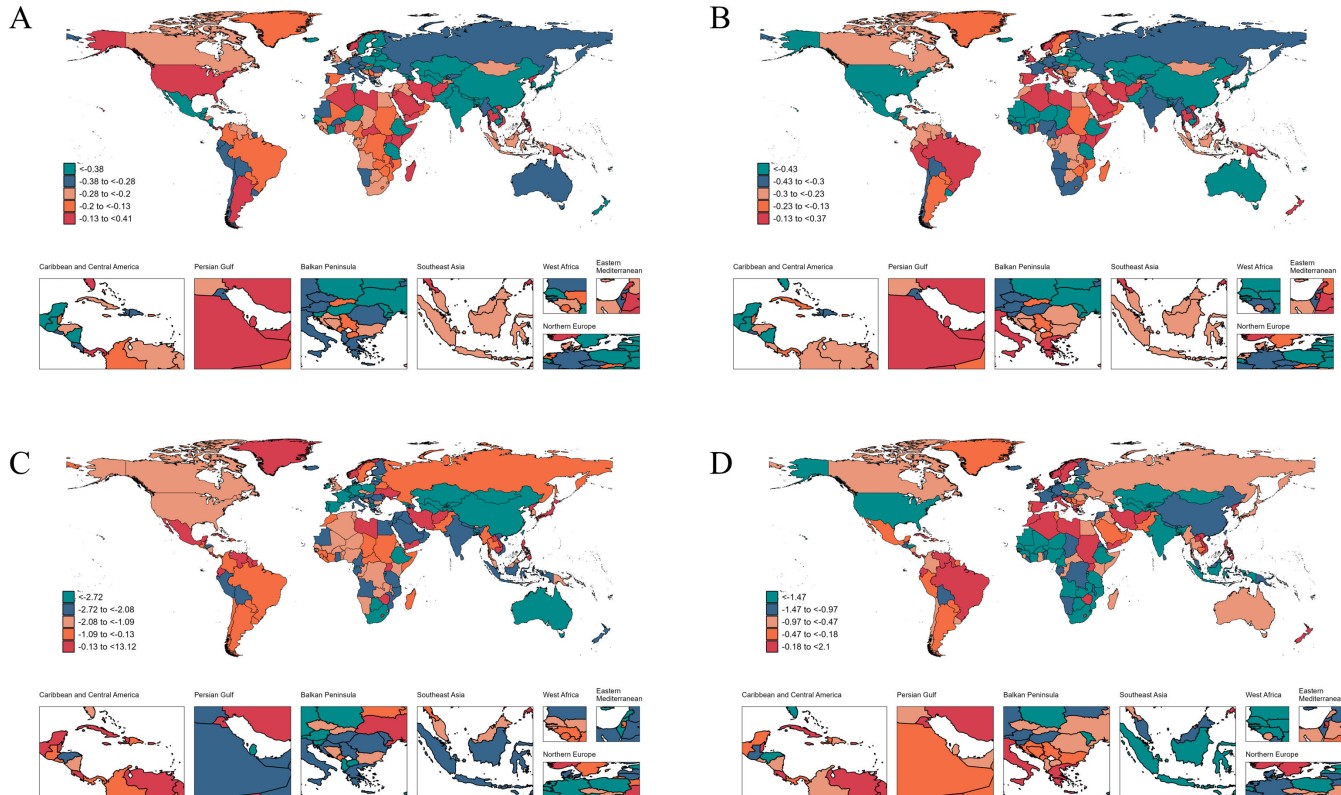

**FIG 4** The average annual percent change in age-standardized incidence (A), prevalence (B), mortality (C), and DALYs (D) rates of gonococcal infection from 1990 to 2021.

(10,974.86 to 31,275.60), and DALYs at 37.49 (13.06 to 84.75) (Table S1). From 1990 to 2021, most regions showed stable or declining trends in these metrics. Western Sub-Saharan Africa showed the most notable increase in age-standardized incidence rates (AAPC: 0.18 [0.03 to 0.32]). However, no upward trends in prevalence or DALY rates were observed across any GBD regions during this period (Table S2).

In 2021, Zimbabwe reported the highest age-standardized incidence rate of trichomoniasis (22,426.64 [13,111.81 to 34,757.76]), while Zambia recorded the highest rates for both prevalence (21,910.47 [12,391.83 to 34,278.10]) and DALYs (41.39 [14.77 to 94.31]). From 1990 to 2021, the most significant decline in the age-standardized incidence rate was observed in Iraq (AAPC: −0.87 [1.06 to −0.67]). Among the other metrics, Burkina Faso showed the most notable decline, with the AAPC of −1.34 (−1.43 to −1.25) for prevalence and −1.36 (−1.44 to −1.28) for DALYs. Conversely, Kenya experienced the most prominent increases across these three metrics, with incidence rising at an AAPC of 0.56 (0.38 to 0.73), prevalence at 0.74 (0.53 to 0.95), and DALYs at 0.72 (0.58 to 0.85) (Table S5; Fig. 5).

## Genital herpes

Globally, in 2021, the estimated age-standardized rates for genital herpes were 1,222.71 (861.48 to 1,650.15) for incidence, 17,137.09 (13,485.32 to 21,121.75) for prevalence, and 4.78 (2.02 to 9.63) for DALYs (Table S1). Between 1990 and 2021, these rates showed increasing trends, with AAPCs 0.24 (0.21 to 0.27) for incidence, 0.18 (0.13 to 0.22) for prevalence, and 0.18 (0.16 to 0.21) for DALYs (Table S2).

Regionally, central Sub-Saharan Africa recorded the highest age-standardized rates in 2021, with incidence at 3,099.41 (2,294.47 to 4,050.97), prevalence at 56,745.62 (47,554.16 to 66,194.76), and DALYs at 0.18 (0.16 to 0.21) (Table S1). From 1990 to 2021, South Asia experienced the largest increases across all metrics, with AAPCs of 0.23 (0.21

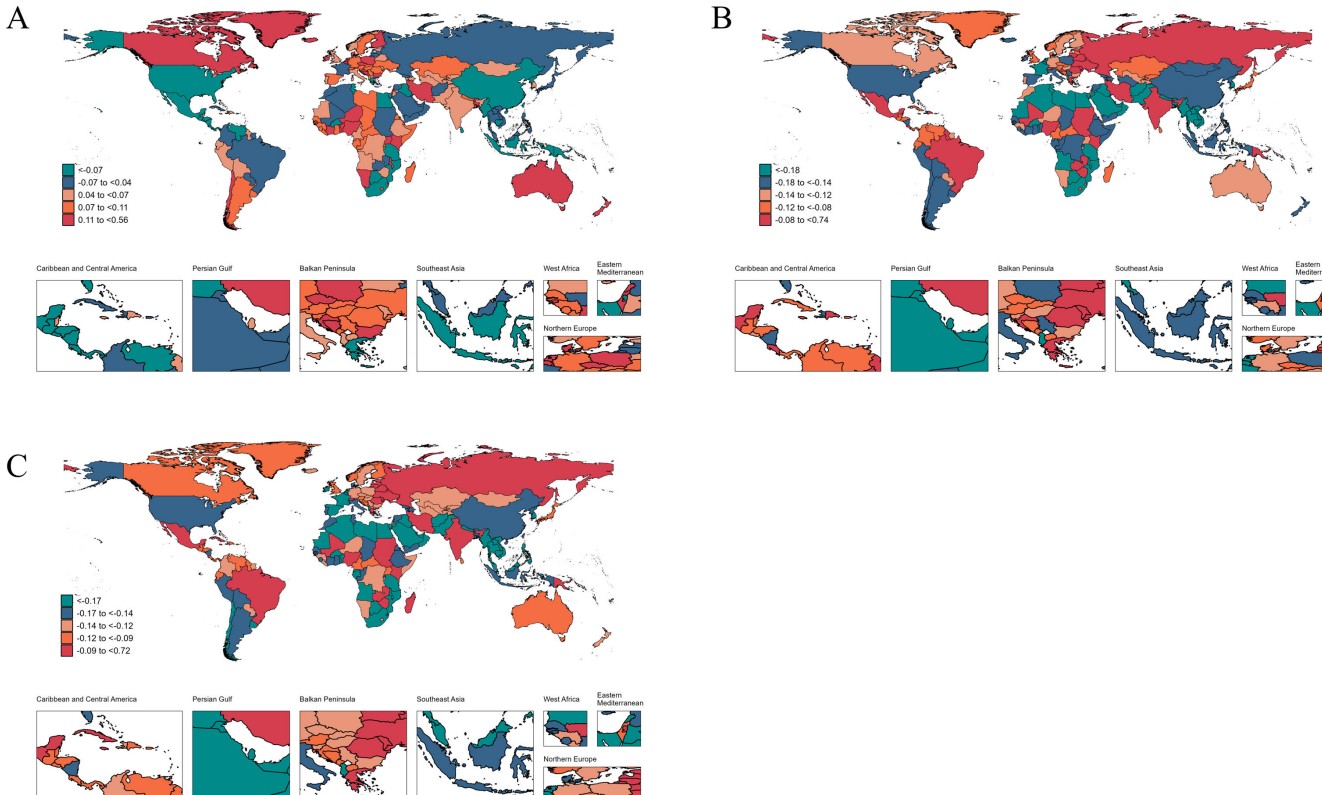

**FIG 5**  The average annual percent change in age-standardized incidence (A); prevalence (B); DALYs (C); rates of trichomoniasis from 1990 to 2021.

to 0.25) for incidence, 0.30 (0.25 to 0.35) for prevalence, and 0.31 (0.27 to 0.35) for DALYs (Table S2).

Equatorial Guinea had the highest age-standardized rates in terms of incidence, prevalence, and DALYs, with values of 3,124.69 (2,304.16 to 4,072.00), 57,776.05 (48,415.58 to 67,132.42), and 15.54 (6.45 to 31.19), respectively. From 1990 to 2021, countries such as Czechia, Eswatini, and India showed the most significant increases in these metrics over the years, with incidence AAPC of 0.53 (0.46 to 0.59) for Czechia, prevalence AAPC of 0.39 (0.37 to 0.40) for Eswatini, and DALY AAPC of 0.40 (0.34 to 0.46) for India. On the other hand, the Republic of Korea exhibited the most notable decline across the three metrics. The AAPCs were as follows: incidence −1.62 (−1.71 to −1.52), prevalence −2.12 (−2.23 to −2.02), mortality −7.18 (−9.01 to −5.3), and DALYs −2.06 (−2.19 to −1.94) (Table S5; Fig. 6).

## SDI regions

Over the past several decades, low-SDI regions have borne the highest burden of STIs in terms of incidence, prevalence, mortality, and DALYs, despite gradual improvements over time. In 2021, the age-standardized rates of incidence (9,418.29 vs 19,719.63), prevalence (21,057.11 vs 46,638.54), mortality (0.54 vs 36.54), and DALYs (54.66 vs 2,326.23) in high- and low-SDI regions, respectively, showed persistent disparities (Fig. 7). From 1990 to 2021, STI metrics generally decreased in low-SDI regions, while incidence and prevalence increased in low-middle SDI regions, and both mortality and DALYs increased in middle-SDI regions and high-middle SDI regions, respectively.

Regarding the distribution of specific STIs, trichomoniasis accounted for the highest proportion of new cases globally, comprising 46.91% of all STI cases, with rates ranging from 43.49% to 60.80% across different SDI regions. However, in high-middle SDI regions, chlamydial infection was the most prevalent infection in terms of new cases, accounting for 44.79% of cases in 2021 (Fig. S1). Genital herpes represented the highest proportion

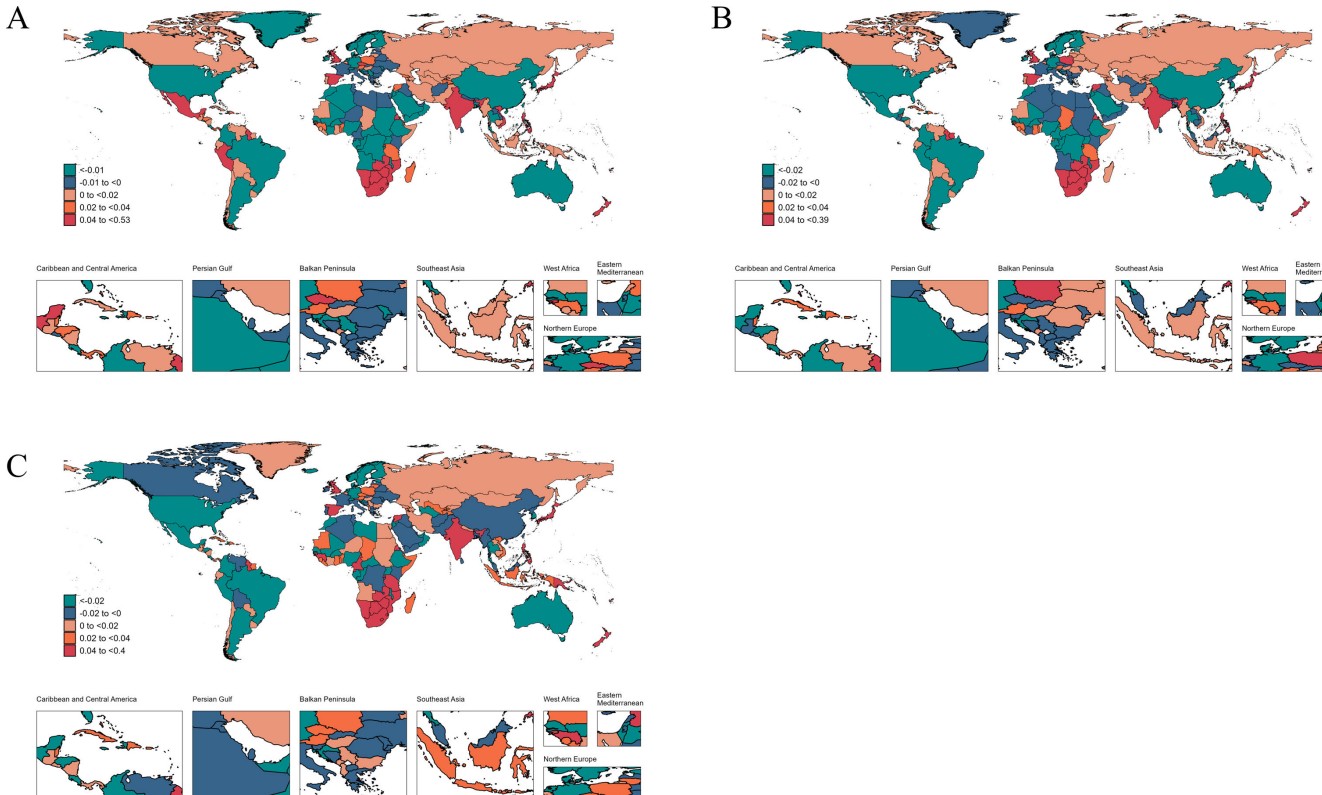

**FIG 6** The average annual percent change in age-standardized incidence (A), prevalence (B), and DALYs (C) rates of genital herpes from 1990 to 2021.

of cases globally (59.25%) and across the five SDI regions, with rates ranging from 56.94% to 68.78%. Additionally, HIV/AIDS accounted for the largest share of deaths and DALYs, both globally and across all SDI regions (Fig. S2). The proportion of deaths attributed to HIV/AIDS ranged from 98.95% to 99.74%, and the proportion of DALYs ranged from 73.14% to 98.17% (Fig. S3 and S4).

## DISCUSSION

This study estimated the age-standardized incidence, prevalence, DALYs, and mortality rates of HIV/AIDS and five other STIs—syphilis, chlamydial infection, gonococcal infection, trichomoniasis, and genital herpes—among women of childbearing age using data from the GBD 2021. The analysis spans from 1990 to 2021, providing a comprehensive overview at global, regional, and national levels, along with an assessment of how these rates have evolved.

Women of childbearing age with STIs face not only the risk of vertical transmission but also an increased likelihood of infertility, gestational hypertension, and adverse pregnancy outcomes, such as preterm delivery (8, 19, 20). While the mortality and DALYs associated with STIs other than HIV/AIDS are generally lower, the incidence and prevalence of these infections remain high globally, warranting continued attention and intervention for women of reproductive age. Between 1990 and 2021, the disease burden of HIV/AIDS, trichomoniasis, and genital herpes showed an upward trend, with the exception of HIV/AIDS incidence rates, which remained stable or decreased. Conversely, syphilis, chlamydia, and gonococcal infections either alleviated or stabilized over the same period.

Among all STIs, genital herpes had the highest prevalence in women of childbearing age. Genital herpes, unlike trichomoniasis, is a leading cause of neonatal transmission, and its management remains a challenge due to the lack of effective treatment to prevent vertical transmission (21, 22). Furthermore, many cases of genital herpes are

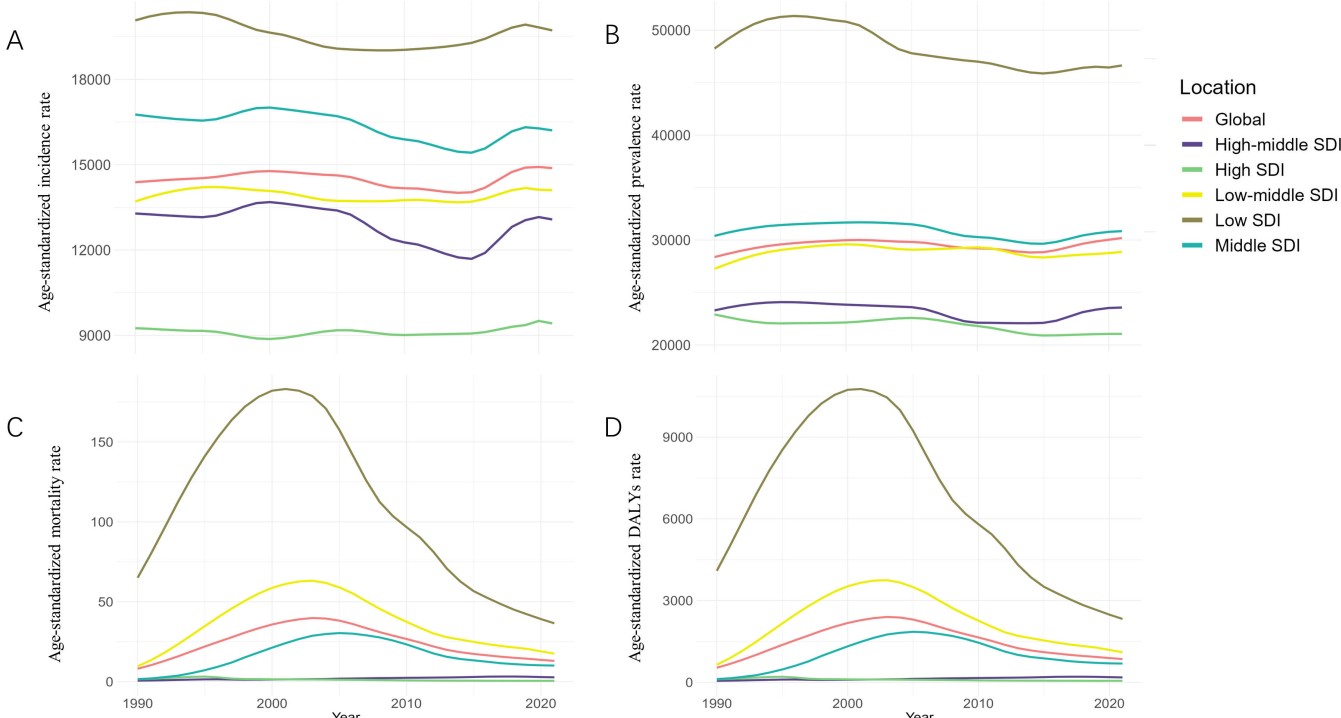

**FIG 7** Age-standardized incidence (A), prevalence (B), mortality (C), and DALYs (D) rates of all STIs in global and five SDI regions from 1990 to 2021.

asymptomatic, complicating efforts at early detection and prevention. As genital herpes is associated with high mortality rates among neonates, its prevention, especially in the context of maternal transmission, remains a significant challenge (23).

By 2021, significant disparities in the burden of STIs are evident across regions, countries, and territories. Low-SDI regions experience a particularly acute disease burden, with many metrics showing increasing trends, consistent with findings in other populations (15, 24). These patterns may stem from limited access to prevention and treatment measures, such as HIV pre-exposure prophylaxis, pregnancy testing, and timely interventions to prevent vertical transmission (25, 26). The prevalence and predominant types of STIs vary across regions, reflecting distinct socioeconomic and healthcare challenges.

Between 1990 and 2021, HIV/AIDS metrics showed significant increases in several regions, particularly in South Asia, where the rising burden may be attributed to factors such as drug abuse, sex trade, and inadequate healthcare infrastructure (27). Conversely, metrics for other STIs, such as gonococcal infection, demonstrated a declining trend in many SDI regions, likely due to improved detection methods and preventive measures (28). Given that early-stage STIs are often asymptomatic, expanding targeted screening programs, particularly in resource-limited regions, is essential for effective disease control (27).

This study provides a comprehensive quantitative analysis of the burden of STIs among women of childbearing age at global, regional, and national levels, addressing a critical gap in prior research. While previous studies often focused on specific regions or individual STIs, this analysis highlights trends in multiple infections over time (29, 30). Different regions need to prioritize prevention and control measures, as well as screening for STIs among women of childbearing age, based on the burden and changing trends of STIs in their respective regions.

This study has some limitations. First, the analysis is limited to six STI diseases, including HIV, and does not encompass the full spectrum of STIs. Second, data on gender minorities were not included, such as transgender and non-binary individuals, as the

GBD 2021 only provided binary sex data (male/female), although these groups may face distinct STI risks. Third, the disease burden in less economically developed regions may have been underestimated due to gaps in healthcare data and infrastructure. High-income countries often utilize high-sensitivity diagnostic modalities, such as PCR testing and genomic sequencing, whereas low-income countries primarily rely on syndromic management or rapid diagnostic tests with limited accuracy. Inadequate testing capacity and restricted access to healthcare services contribute to a high proportion of undiagnosed and unreported cases (31). Moreover, sociocultural factors further exacerbate these surveillance gaps, including limited sexual health education, persistent stigma surrounding STIs, and structural gender inequalities that hinder healthcare-seeking behavior (32). Finally, the term "global rates" refers to estimates based on the 204 countries and territories covered by the GBD 2021, which, although broadly representative, may not include every region of the world.

In conclusion, this study provides age-standardized rates and trends for several common STIs among women of reproductive age from 1990 to 2021. The findings reveal substantial regional variation in the burden of these infections, underscoring the need for targeted screening strategies tailored to local conditions. Enhanced prevention and control efforts, particularly in resource-limited areas, are critical to mitigating the spread and impact of STIs globally.

## ACKNOWLEDGMENTS

This work was supported by the National Natural Science Foundation of China (No. 82072134), Excellent Young Talents Fund Program of Higher Education Institutions of Anhui Province (No. gxyqZD2018026) in 2018, and a grant from the Anhui Medical University Fund (No. 2021xkj177).

## AUTHOR AFFILIATIONS

[1]The Second Department of Critical Care Medicine, The Second Affiliated Hospital of Anhui Medical University, Hefei, Anhui, China
[2]Laboratory of Cardiopulmonary Resuscitation and Critical Care, The Second Affiliated Hospital of Anhui Medical University, Hefei, Anhui, China

## AUTHOR ORCIDs

Xiaoyu Zhang  http://orcid.org/0000-0002-7725-3669
Min Yang  http://orcid.org/0000-0002-3484-2447

## FUNDING

| Funder | Grant(s) | Author(s) |
| --- | --- | --- |
| National Natural Science Foundation of China | 82072134 | Min Yang |
| Excellent Young Talents Fund Program of Higher Education Institutions of Anhui Province | gxyqZD2018026 | Min Yang |
| Anhui Medical University | 2021xkj177 | Hui Li |

## AUTHOR CONTRIBUTIONS

Xiaoyu Zhang, Investigation, Methodology, Visualization, Writing – original draft | Chenglong Hu, Data curation, Methodology, Software, Writing – original draft | Yao Liang, Methodology, Validation, Visualization | Wanguo Dong, Data curation, Software | Jian Gao, Methodology, Visualization | Yu Ji, Data curation, Methodology | Chang Cao, Validation, Visualization | Wei Shi, Data curation, Investigation | Shuaijie Zhu, Writing – original draft | Heng Guo, Software, Visualization | Tianfeng Hua, Supervision, Writing – review and editing | Hui Li, Funding acquisition, Supervision | Min Yang, Funding acquisition, Supervision, Writing – review and editing

## DATA AVAILABILITY

The data underlying this article are available in Global Burden of Diseases 2021, at https://vizhub.healthdata.org/gbd-results/. The base map data supporting this study are available from the Resource and Environmental Sciences Data Center, Chinese Academy of Sciences (RESDC) at https://www.resdc.cn/. These maps are for illustrative purposes only and do not represent official administrative boundaries.

## ADDITIONAL FILES

The following material is available online.

### Supplemental Material

**Supplemental figures (Spectrum00488-25-s0001.docx).** Fig. S1 to S4.
**Table S1 (Spectrum00488-25-s0002.docx).** The age-standardized incidence, prevalence, mortality, and DALY rates at global and regional levels in 2021.
**Table S2 (Spectrum00488-25-s0003.docx).** The change trends of age-standardized rates from 1990 to 2021 in the global and regional levels.
**Table S3 (Spectrum00488-25-s0004.docx).** The age-standardized incidence, prevalence, mortality, and DALY rates of 204 countries and territories.
**Table S4 (Spectrum00488-25-s0005.docx).** The age-standardized incidence, prevalence, mortality, and DALY rates of 204 countries and territories.
**Table S5 (Spectrum00488-25-s0006.docx).** The age-standardized incidence, prevalence, mortality, and DALY rates of 204 countries and territories.

### Open Peer Review

**PEER REVIEW HISTORY (review-history.pdf).** An accounting of the reviewer comments and feedback.

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
