## [Reviewer comments · Microbiology Spectrum]

Microbiology Spectrum

Global, regional and national burden of HIV and other sexually transmitted infections among women of childbearing age from 1990 to 2021

Xiaoyu Zhang, Chenglong Hu, Yao Liang, Wanguo Dong, Jian Gao, Yu Ji, Chang Cao, Wei Shi, Shuaijie Zhu, Heng Guo, Tianfeng Hua, Hui Li, and Min Yang

Corresponding Author(s): Min Yang, Second Affiliated Hospital of Anhui Medical University

Review Timeline:

Submission Date:	February 19, 2025
Editorial Decision:	June 9, 2025
Revision Received:	August 11, 2025
Accepted:	September 23, 2025

Editor: Ayesha Khan

Reviewer(s): Disclosure of reviewer identity is with reference to reviewer comments included in decision letter(s). The following individuals involved in review of your submission have agreed to reveal their identity: Filipe Cerqueira (Reviewer #1); Anatasia Weiland (Reviewer #2)

Transaction Report:

DOI: <https://doi.org/10.1128/spectrum.00488-25>

Re: Spectrum00488-25 (Global, regional and national burden of HIV and other sexually transmitted infections among women of childbearing age from 1990 to 2021)

Dear Dr. Min Yang:

Thank you for the privilege of reviewing your work. Below you will find my comments, instructions from the Spectrum editorial office, and the reviewer comments.

Revision Guidelines

Sincerely,
Ayesha Khan
Editor
Microbiology Spectrum

Reviewer #1 (Comments for the Author):

Zhang et al presents an important report of the global, regional and national burden of STIs and HIV infection among women of childbearing age from 1990-2021. Using the Global Burden of Disease Database, the authors analyzed relevant data. These data can inform policy makers on strategic resource allocation. Please see my notes for authors to address below:
Paragraph starting in Line 367 needs to be expanded. Please elaborate on "gaps in health data and infrastructure." How can

access to testing and different STI testing modalities impact the conclusions reached from this study? How about different social norms in the countries mentioned?

Line 109 states "studies" (plural) but only one citation was provided. Please include more studies that use this estimation method.

Figures 1-6 are nearly illegible. They are deformed and of low resolution

Reviewer #2 (Comments for the Author):

The authors provide comprehensive analyses of publicly available dataset on sexually transmitted diseases from 1990-2021. The result of their work help scientific community understand global trends in sexually transmitted diseases among women of childbearing age. The authors additionally provide their conclusions on factors associated with STI transmission in this population, and the ways to address disease burden.

Manuscript #: Spectrum00488-25

Title: Global, regional and national burden of HIV and other sexually transmitted infections among women of childbearing age from 1990 to 2021

Major comments:

- 1) Lowest APPCs by country reported for trichomoniasis and gonococcal infection, but not for the rest of the infections. Consider including for all STIs for consistency. This would be particularly appreciated for infections with decline in APPCs.

- 2) Abstract, Lines 35-39:
 - a. The authors provide the highest rates for incidence, mortality, prevalence, and DALYs for different STIs in 2021. It is unclear what authors compare these rates to, as the “highest” is a relative measure. Do authors compare year-to-year, i.e., year 2021 to all reviewed individual years (1990, 1991, 1992, 1993, etc...2021) , entire timeframe (1990-2021), or do they compare rates of one STI to another STI for that particular year (2021)? Please clarify.

 - b. In this paper, the authors reviewed the trends over 31 years, however, this sentence provides insight on 2021 year only. It would be appreciated if the authors included their conclusions on STIs trends based on APPCs over 1990-2021 instead, as this measure provides a much clearer understanding of diseases’ burden over time, than the absolute numbers for a particular year.

- 3) Results, Lines 167-170: The authors report a single value for incidence and other measures for several countries combined, example: 1506.26 (792.30 to 2490.52) for Equatorial Guinea, Liberia, South Sudan, and the Central African Republic. How was this value calculated? Are these values averaged among all listed countries? These values do not correspond to the data in Table S1 that is referenced.

- 4) Results, Lines 172-175, Lines 289-292, same comment (#3) applies as mentioned above.

Minor comments:

- 1) Consider different word choice instead of “indicators”, for example, “metrics” throughout the manuscript, to describe disease incidence, prevalence, etc.

- 2) Mortality rates for trichomonas and genital herpes are not reported. Explanation (assuming rates are close to 0 for those infections) on why these rates were omitted is appreciated.
- 3) Importance, Lines 49, 53 – unnecessary spelling of SSTI and DALY acronyms, as these were already included in the abstract
- 4) Introduction, Line 63 – no need to spell out STIs again
- 5) Introduction, Line 66-67: Women of childbearing age, especially those who are sexually active, are at increased risk of these infections due to their unique anatomical features.” Clarification on what authors mean by unique anatomical features is needed, as well as a supporting reference.
- 6) Introduction, Line 70-71: “Infertility, often a consequence of STIs, can also contribute to unsafe sexual behaviors, thereby increasing the risk of further STI transmission 11,12” – While this association is not commonly studied and reported in literature, it is an important point that authors decided to include. This statement, however, is only supported by the reference #1, Dhont et al 2011 (and for the HIV transmission only, not for other STIs), not #11.
- 7) Introduction, Line 74-75, citation needed.
- 8) Data analysis, Line 110 – APPC acronym has already been spelled out.
- 9) Line 279-280: add a break line between two paragraphs
- 10) Trichomoniasis, Line 248, 262 – trichomoniasis should not be capitalized
- 11) SDIs Regions, Line 305-307: “From 1990 to 2021, STI indicators generally decreased in low-SDI regions, while incidence and prevalence increased in low-middle SDI regions, and both mortality and DALYs increased in middle-SDI regions and high-middle SDI regions, respectively.” – consider revising as the first part of this sentence contradict the previous sentence: “Over the past several decades, high-SDI regions have experienced minimal burden from STIs, while low-SDI areas have borne the highest burden”
- 12) Figure 7 – consider adding “... cumulative rates of all STIs” to the title.

- 13) Discussion, Line 338-339: it is unclear what is considered the “general population” that women of childbearing age are compared to in this sentence.
- 14) Discussion, Line 351-352: “The <...> focus of various STIs” - consider revising.
- 15) Discussion, Lines 353-355: consider revising into one sentence instead of two.
- 16) Discussion, Line 368: consider clarifying “sexual minorities”.
- 17) Discussion, Limitations: consider adding a sentence that the analysis does not include all countries worldwide, despite “global rates” reported throughout the manuscript, as the authors used the dataset from 204 countries.

Reviewer #1 (Comments for the Author):

Zhang et al presents an important report of the global, regional and national burden of STIs and HIV infection among women of childbearing age from 1990-2021. Using the Global Burden of Disease Database, the authors analyzed relevant data. These data can inform policy makers on strategic resource allocation. Please see my notes for authors to address below:

1. Paragraph starting in Line 367 needs to be expanded. Please elaborate on "gaps in health data and infrastructure." How can access to testing and different STI testing modalities impact the conclusions reached from this study? How about different social norms in the countries mentioned?

-Thank you for the valuable suggestion. We have revised the manuscript accordingly in the Limitations section (lines 391–399) to clarify the differences in diagnostic approaches and to address the influence of sociocultural factors on STI surveillance

2. Line 109 states "studies" (plural) but only one citation was provided. Please include more studies that use this estimation method.

-Thank you for pointing this out. We have added additional relevant references in line 110 to support the use of this estimation method.

3. Figures 1-6 are nearly illegible. They are deformed and of low resolution

-Thank you for your comment. We have replaced Figures 1–6 with high-resolution versions to improve clarity and readability.

Reviewer #2

Major comments:

1) Lowest APPCs by country reported for trichomoniasis and gonococcal infection, but not for the rest of the infections. Consider including for all STIs for consistency. This would be particularly appreciated for infections with decline in APPCs.

-Thank you for the suggestion. We have now added the lowest AAPCs by country for all STIs to ensure consistency across the results.

2) Abstract, Lines 35-39:

a. The authors provide the highest rates for incidence, mortality, prevalence, and DALYs for different STIs in 2021. It is unclear what authors compare these rates to, as the “highest” is a relative measure. Do authors compare year-to-year, i.e., year 2021 to all reviewed individual years (1990, 1991, 1992, 1993, etc...2021), entire timeframe (1990-2021), or do they compare rates of one STI to another STI for that particular year (2021)? Please clarify.

-Thank you for the helpful comment. We have clarified in the revised text that the “highest” rates refer to comparisons among different STIs in the year 2021 (line 35).

b. In this paper, the authors reviewed the trends over 31 years, however, this sentence provides insight on 2021 year only. It would be appreciated if the authors included their conclusions on STIs trends based on APCs over 1990-2021 instead, as this measure provides a much clearer understanding of diseases' burden over time, than the absolute numbers for a particular year.

-Thank you for the suggestion. We have revised the text to include conclusions on long-term STI trends based on APCs from 1990 to 2021 (lines 39-44).

3) Results, Lines 167-170: The authors report a single value for incidence and other measures for several countries combined, example: 1506.26 (792.30 to 2490.52) for Equatorial Guinea, Liberia, South Sudan, and the Central African Republic. How was this value calculated? Are these values averaged among all listed countries? These values do not correspond to the data in Table S1 that is referenced.

-Thank you for the helpful comment. We have revised the text to report the highest value for each indicator along with its corresponding country, and clarified that no combined or averaged values were used. Additionally, the original reference to Table S1 was a typographical error and has been corrected to Table S3 (lines 170-173).

4) Results, Lines 172-175, Lines 289-292, same comment (#3) applies as mentioned above.

-As with the previous comment, we have revised these sections to clearly present the highest value for each indicator along with its specific country (lines 173-177, lines 302-305).

Minor comments:

1) Consider different word choice instead of “indicators”, for example, “metrics” throughout the manuscript, to describe disease incidence, prevalence, etc.

-Thank you for the suggestion. We have replaced “indicators” with “metrics” throughout the manuscript where appropriate to improve clarity and consistency.

2) Mortality rates for trichomonas and genital herpes are not reported. Explanation (assuming rates are close to 0 for those infections) on why these rates were omitted is appreciated.

-Thank you for the comment. Mortality rates for trichomonas and genital herpes were not reported because such data were not available in the GBD 2021 dataset. A brief explanation has been added in the Results section to clarify this point in the table footnote.

3) Importance, Lines 49, 53 – unnecessary spelling of SSTI and DALY acronyms, as these were already included in the abstract

-Thank you for pointing this out. We have removed the repeated spelling of the STI and DALY acronyms in the Importance section.

4) Introduction, Line 63 – no need to spell out STIs again

-Thank you for the suggestion. We have removed the repeated spelling of STIs in line 64.

5) Introduction, Line 66-67: Women of childbearing age, especially those who are sexually active, are at increased risk of these infections due to their unique anatomical features.”

Clarification on what authors mean by unique anatomical features is needed, as well as a supporting reference.

-Thank you for the comment. We have revised the sentence in lines 69–71 to clarify the anatomical factors contributing to increased STI risk among women of childbearing age and have added supporting references.

6) Introduction, Line 70-71: “Infertility, often a consequence of STIs, can also contribute to unsafe sexual behaviors, thereby increasing the risk of further STI transmission 11,12” –

While this association is not commonly studied and reported in literature, it is an important point that authors decided to include. This statement, however, is only supported by the reference #1, Dhont et al 2011 (and for the HIV transmission only, not for other STIs), not #11.

-Thank you for the comment. We have removed the sentence to avoid overinterpretation and ensure accurate referencing.

7)Introduction, Line 74-75, citation needed.

-Thank you for the suggestion. We have added the appropriate citation to support the statement in lines 76.

8)Data analysis, Line 110 – APPC acronym has already been spelled out.

-Thank you for pointing this out. We have removed the repeated spelling of the AAPC acronym in line 111.

9)Line 279-280: add a break line between two paragraphs

-Thank you for the suggestion. We have added a paragraph break between lines 293 and 294 as requested.

10)Trichomoniasis, Line 248, 262 – trichomoniasis should not be capitalized

-Thank you for pointing this out. We have corrected the capitalization of trichomoniasis.

11) SDIs Regions, Line 305-307: “From 1990 to 2021, STI indicators generally decreased in

low-SDI regions, while incidence and prevalence increased in low-middle SDI regions, and both mortality and DALYs increased in middle-SDI regions and high-middle SDI regions, respectively.” – consider revising as the first part of this sentence contradict the previous sentence: “Over the past several decades, high-SDI regions have experienced minimal burden from STIs, while low-SDI areas have borne the highest burden”

-Thank you for the insightful comment. We have revised the sentence to clarify that although low-SDI regions have the highest burden, a declining trend was observed over time, thereby eliminating the contradiction (lines 316-320).

12) Figure 7 – consider adding “... cumulative rates of all STIs” to the title.

-Thank you for the suggestion. We have revised the title of Figure 7 to specify that it presents cumulative rates of all STIs.

13) Discussion, Line 338-339: it is unclear what is considered the “general population” that women of childbearing age are compared to in this sentence.

-Thank you for pointing this out. Upon review, we found that the original comparison was inaccurate and have revised the sentence to simply state that genital herpes had the highest prevalence among STIs (line 356).

14) Discussion, Line 351-352: “The <...> focus of various STIs” - consider revising.

-Thank you for the suggestion. We have revised the wording to improve clarity and replaced “focus” with more appropriate terminology (lines 368-370).

15) Discussion, Lines 353-355: consider revising into one sentence instead of two.

-Thank you for the comment. We have revised the text and combined the two sentences into one for improved flow and conciseness.

16) Discussion, Line 368: consider clarifying “sexual minorities”.

-Thank you for the suggestion. We have clarified the term “sexual minorities” by specifying examples such as transgender and non-binary individuals (lines 388-390).

17) Discussion, Limitations: consider adding a sentence that the analysis does not include all countries worldwide, despite “global rates” reported throughout the manuscript, as the authors used the dataset from 204 countries.

-Thank you for the helpful comment. We have added a sentence in the Limitations section to clarify that the reported global rates are based on data from 204 countries and territories included in the GBD 2021 dataset (lines 398-400).

Re: Spectrum00488-25R1 (Global, regional and national burden of HIV and other sexually transmitted infections among women of childbearing age from 1990 to 2021)

Dear Dr. Min Yang:

Your manuscript has been accepted, and I am forwarding it to the ASM production staff for publication. Your paper will first be checked to make sure all elements meet the technical requirements. ASM staff will contact you if anything needs to be revised before copyediting and production can begin. Otherwise, you will be notified when your proofs are ready to be viewed.

Sincerely,
Ayesha Khan
Editor
Microbiology Spectrum